# On the Vulnerability of Adversarially Trained Models Against Two-faced Attacks

**Shengjie Zhou**[1]    **Lue Tao**[2]    **Yuzhou Cao**[3]    **Tao Xiang**[1]    **Bo An**[4,3]    **Lei Feng**[3*]
[1]Chongqing University    [2]Nanjing University
[3]Nanyang Technological University    [4]Skywork AI
zshengjie@cqu.edu.cn, taol@lamda.nju.edu.cn, txiang@cqu.edu.cn
yuzhou002@e.ntu.edu.sg, boan@ntu.edu.sg, lfengqaq@gmail.com

## Abstract

Adversarial robustness is an important standard for measuring the quality of learned models, and adversarial training is an effective strategy for improving the adversarial robustness of models. In this paper, we disclose that adversarially trained models are vulnerable to *two-faced attacks*, where slight perturbations in input features are crafted to make the model exhibit a false sense of robustness in the verification phase. Such a threat is significantly important as it can mislead our evaluation of the adversarial robustness of models, which could cause unpredictable security issues when deploying substandard models in reality. More seriously, this threat seems to be pervasive and tricky: we find that *many types of models suffer from this threat, and models with higher adversarial robustness tend to be more vulnerable*. Furthermore, we provide the first attempt to formulate this threat, disclose its relationships with adversarial risk, and try to circumvent it via a simple countermeasure. These findings serve as a crucial reminder for practitioners to exercise caution in the verification phase, urging them to refrain from blindly trusting the exhibited adversarial robustness of models.

## 1 Introduction

Adversarial robustness plays a crucial role in evaluating modern machine learning models (Paterson et al., 2021; Hutchinson et al., 2020), especially in safety-critical tasks like autonomous driving (Feng et al., 2021). It is well-known that naturally trained models usually have low accuracy on adversarial examples (Szegedy et al., 2014; Biggio et al., 2013), thereby being hard to exhibit sufficient robustness required for passing the verification phase (Paterson et al., 2021). To mitigate this issue, a widely adopted technique is adversarial training, which enhances the adversarial robustness of models by incorporating adversarial examples into training (Goodfellow et al., 2015; Madry et al., 2018). This enables the model to adapt and withstand potential verification-time adversarial examples (Athalye et al., 2018). Recently, adversarial training was further shown to be capable of mitigating the threat of training-time availability attacks (Huang et al., 2021; Tao et al., 2021).

However, adversarial training may not be all you need for constructing robust models. A recent study showed that adversarial training may fail to provide adversarial robustness in the verification phase when the provided training data is manipulated by training-time stability attacks (Tao et al., 2022a), which discloses the vulnerability of adversarial training under such a training-time threat. It is worth noting that to enhance the model deployment success rate, it is necessary to conduct model verification and long-term monitoring (Paleyes et al., 2022). This motivates us to consider whether there exists a verification-time threat that can fool the adversarially trained models by slightly perturbing the provided data in the verification phase.

In this work, we for the first time show that adversarially trained models can be vulnerable to a new type of verification-time threat called *two-faced attacks*. Two-faced attacks focus on the robustness of the validation phase in the machine learning workflow (Paterson et al., 2021) shown in Figure 1(a). In contrast to the commonly known adversarial examples that deliberately induce models to make

---

*Corresponding author.

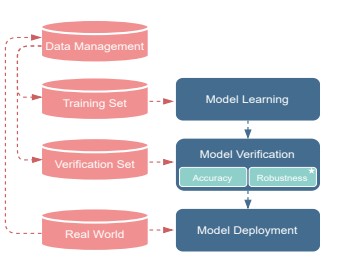 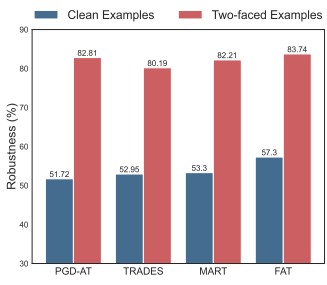 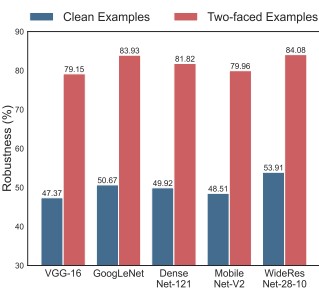

(a) Machine learning workflow.  (b) Varying AT methods.  (c) Varying network architectures.

Figure 1: (a) Machine learning workflow, we focus on the robustness of the verification phase; (b) ResNet-18 trained by PGD-AT, TRADES, MART, and FAT on CIFAR-10; (c) Different model architectures trained by PGD-AT on CIFAR-10.

incorrect predictions, two-faced attacks aim to give a false sense of high adversarial robustness of (substandard) adversarially trained models. For example, as shown in Figure 1(b), ResNet-18 (He et al., 2015) trained by PGD-AT on CIFAR-10 (Krizhevsky, 2009) has the actual verification robustness (i.e., adversarial accuracy) of only 51.72%, while it becomes 82.81% after two-faced attacks. Moreover, as post-deployment monitoring metrics overlap with validation metrics (Paleyes et al., 2022), researchers relying on these indicators may choose not to modify the model, leading to a series of security issues. Figure 1(c) shows that robust models with different architectures are vulnerable to two-faced attacks. Therefore, two-faced attacks are significantly important and cannot be simply ignored before model deployment in safety-critical applications.

To provide a comprehensive study on two-faced attacks, we verify the adversarial robustness of various network architectures (e.g., ResNet-18 and WideResNet-28-10) on multiple benchmark datasets including CIFAR-10, SVHN (Netzer et al., 2011), CIFAR-100 (Krizhevsky, 2009), and Tiny-ImageNet (Yao et al., 2015), employing different adversarial training methods including PGD-AT (Madry et al., 2018), TRADES (Zhang et al., 2019), MART (Wang et al., 2020b), FAT (Zhang et al., 2020), and THRM (Tao et al., 2022b). In addition, we validate various off-the-shelf adversarially trained models from RobustBench (Croce et al., 2021), with diverse architectures and training strategies. We also evaluate the threat of two-faced attacks against different robustness verification methords and examine the transferability of two-faced attacks. All experimental results consistently show that these models exhibit higher robustness under two-faced attacks compared with their actual robustness, which demonstrates that adversarially trained models are indeed vulnerable to two-faced attacks, and such attacks may be widespread.

Furthermore, we introduce the concept of *two-faced risk* for the first time to understand two-faced attacks theoretically and empirically. Specifically, we theoretically establish the relationship between the two-faced risk and the adversarial risk. We also empirically quantify the two-faced risk of adversarially trained models under varying levels of adversarial risk. Our results present an intriguing trade-off between the two-faced risk and the adversarial risk: *models that are more robust against adversarial examples tend to be more vulnerable to two-faced attacks*. This indicates an inherent difficulty in defending against the threat of two-faced attacks.

In summary, our study reveals the threat of two-faced attacks in the model verification phase, which may be widespread in adversarially trained models. These findings serve as a crucial reminder for practitioners to exercise caution during the validation phase, urging them to refrain from blindly trusting the exhibited adversarial robustness of models, otherwise an unqualified model may cause serious consequences when deployed in safety-critical scenarios.

## 2 PRELIMINARIES

In this section, we formally introduce two types of evasion attacks (Biggio et al., 2013), namely adversarial attacks (Szegedy et al., 2014) and hypocritical attacks (Tao et al., 2022b). It is worth noting that some training-time poisoning attacks (e.g., unlearnable examples (Huang et al., 2021; Fu

et al., 2022; Tao et al., 2021)) are also related to our work to some degree, but we do not introduce these poisoning attacks since our work focuses on verification-time evasion attacks.

**Setup.** We consider a classification task with the input data $(\boldsymbol{x}, y) \in \mathbb{R}^d \times [K]$ is sampled from an underlying distribution $\mathcal{D}$. The goal is to train a DNN classifier $f_{\boldsymbol{\theta}} : \mathbb{R}^d \rightarrow [K]$ that can accurately predict the label $y$ given an input $\boldsymbol{x}$.

**Adversarial Attacks.** Adversarial attacks refer to the creation of adversarial examples by an *adversary* to mislead the model into making incorrect predictions. Given a classifier $f_{\boldsymbol{\theta}}$ and an input example $(\boldsymbol{x}, y)$, the $\epsilon$-bounded adversarial instance $\boldsymbol{x}_{\text{adv}}$ is defined as:

$$\boldsymbol{x}_{\text{adv}} = \boldsymbol{x} + \arg\max_{\|\boldsymbol{\delta}\| \leq \epsilon} \mathbb{1}\left(f_{\boldsymbol{\theta}}\left(\boldsymbol{x} + \boldsymbol{\delta}\right) \neq y\right), \tag{1}$$

where $\|\cdot\|$ represents the norm (e.g., the $\ell_2$ or $\ell_\infty$ norm), and $\mathbb{1}(\cdot)$ denotes the indicator function that returns 1 if the argument is true otherwise 0.

**Adversarial Risk.** Since adversarial examples are extremely threatening to the model, the defender needs to train a model with high robustness to effectively mitigate this threat. The primary objective is to train a model that has a low *adversarial risk* under the defense budget $\epsilon$:

$$\mathcal{R}_{\text{adv}}\left(f_{\boldsymbol{\theta}}, \mathcal{D}\right) = \mathbb{E}_{(\boldsymbol{x}, y) \sim \mathcal{D}}\left[\max_{\|\boldsymbol{x}' - \boldsymbol{x}\| \leq \epsilon} \mathbb{1}\left(f_{\boldsymbol{\theta}}\left(\boldsymbol{x}'\right) \neq y\right)\right]. \tag{2}$$

Adversarial training (Madry et al., 2018; Athalye et al., 2018) is a widely adopted paradigm to accomplish this objective. Popular adversarial training methods include PGD-AT (Madry et al., 2018), TRADES (Zhang et al., 2019), and MART (Wang et al., 2020b).

**Hypocritical Attacks.** Hypocritical attacks (Tao et al., 2022b) refer to the creation of hypocritical examples by a *false friend* to induce the model to correctly classify originally misclassified inputs. Hypocritical attacks perturb a misclassified input $(\boldsymbol{x}, y)$ so that the model can classify it correctly. Given a classifier $f_{\boldsymbol{\theta}}$ and an input $(\boldsymbol{x}, y)$, the $\epsilon$-bounded hypocritical example $\boldsymbol{x}_{\text{hyp}}$ is defined as:

$$\boldsymbol{x}_{\text{hyp}} = \boldsymbol{x} + \arg\max_{\|\boldsymbol{\delta}\| \leq \epsilon} \mathbb{1}\left(f_{\boldsymbol{\theta}}\left(\boldsymbol{x} + \boldsymbol{\delta}\right) = y\right). \tag{3}$$

It was shown that hypocritical attacks can make the model exhibit extremely high verification accuracy, thereby concealing the deficiencies of the model. Adversarial training can mitigate hypocritical attacks to some degree, but these attacks are still in need of special countermeasures.

## 3 STUDY ON TWO-FACED ATTACKS

In this section, we formally introduce the two-faced attacks studied in this paper, which aim to make the model have a false sense of high adversarial robustness.

### 3.1 FORMULATION OF TWO-FACED EXAMPLES

**Two-Faced Attacks.** The aforementioned two types of attacks focus on improving or reducing the accuracy of a model, while two-faced attacks focus on deceptively enhancing the adversarial accuracy, i.e., adding a perturbation $\boldsymbol{\delta}$ to data so that it can still be correctly classified in various adversarial verification algorithms. Given a classifier $f_{\boldsymbol{\theta}}$ and an input $(\boldsymbol{x}, y)$, the $\epsilon$-bounded two-faced example $\boldsymbol{x}_{\text{tf}}$ is defined as follows:

$$\boldsymbol{x}_{\text{tf}} = \boldsymbol{x} + \arg\min_{\|\boldsymbol{\delta}\| \leq \epsilon}\left[\max_{\|\boldsymbol{t}\| \leq \epsilon} \mathbb{1}\left(f_{\boldsymbol{\theta}}\left(\boldsymbol{x} + \boldsymbol{\delta} + \boldsymbol{t}\right) \neq y\right)\right]. \tag{4}$$

For the practical implementation of Eq. (4), we can replace the indicator function with the commonly used cross-entropy loss function ($\ell_{\text{CE}}$) and optimize it using the projected gradient descent (PGD) method. It is worth noting that Eq. (4) poses a challenging bi-level optimization problem for training a DNN classifier. To approximate the solution, we employ the following optimization procedure: we first calculate the inner perturbation $\boldsymbol{t}^* = \arg\max_{\|\boldsymbol{t}\| \leq \epsilon}\ell_{\text{CE}}(f_{\boldsymbol{\theta}}(\boldsymbol{x} + \boldsymbol{\delta}^* + \boldsymbol{t}), y)$ via PGD, and then calculate the outer perturbation $\boldsymbol{\delta}^* = \arg\min_{\|\boldsymbol{\delta}\| \leq \epsilon}\ell_{\text{CE}}(f_{\boldsymbol{\theta}}(\boldsymbol{x} + \boldsymbol{\delta} + \boldsymbol{t}^*), y)$ via PGD. The two procedures are repeated in an alternating way. The detailed process is presented in Algorithm 1.

---

**Algorithm 1** Two-Faced Examples Generation

---

**Input:** Model $f_{\boldsymbol{\theta}}$, Sample $(\boldsymbol{x}, y)$, Number of Iterations $N$
**Output:** Optimized perturbation $\boldsymbol{x}_{\text{tf}}$
1: Initialize the perturbation $\boldsymbol{\delta}^*$
2: **for** $n = 1$ to $N$ **do**
3:     Find the optimal perturbation $\boldsymbol{t}^* = \arg\max_{\|\boldsymbol{t}\| \leq \epsilon} \ell_{\text{CE}}(f_{\boldsymbol{\theta}}(\boldsymbol{x} + \boldsymbol{\delta}^* + \boldsymbol{t}), y)$
4:     Find the optimal perturbation $\boldsymbol{\delta}^* = \arg\min_{\|\boldsymbol{\delta}\| \leq \epsilon} \ell_{\text{CE}}(f_{\boldsymbol{\theta}}(\boldsymbol{x} + \boldsymbol{\delta} + \boldsymbol{t}^*), y)$
5: **end for**
6: Calculate the perturbed instance $\boldsymbol{x}_{\text{tf}} = \boldsymbol{x} + \boldsymbol{\delta}^*$
7: **Return** $\boldsymbol{x}_{\text{tf}}$

---

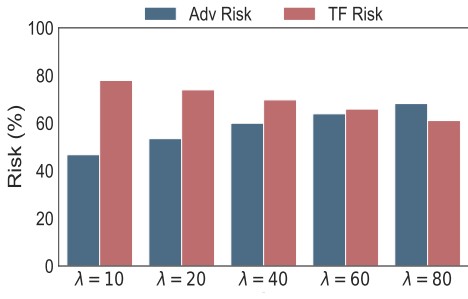
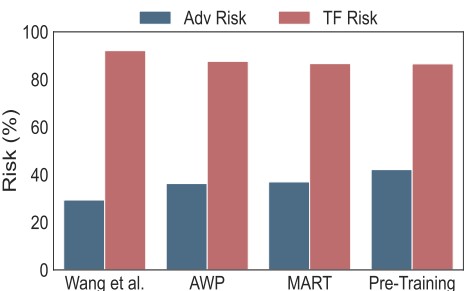

(a) Different values of $\lambda$ in TRADES.      (b) Different AT models from RobustBench.

Figure 2: (a) Adversarial risk and two-faced risk of ResNet-18 under different $\lambda$ on CIFAR-10; (b) Adversarial risk and two-faced risk of WideResNet-28-10 under different RobustBench models on CIFAR-10: Wang et al. (2023), AWP (Wu et al., 2020), MART (Wang et al., 2020b), and Pre-Training (Hendrycks et al., 2019).

### 3.2 FORMULATION OF TWO-FACED RISK

**Two-Faced Risk.** To provide a better understanding of the threat posed by two-faced attacks, we introduce the concept of two-faced risk. Given a classifier $f_{\boldsymbol{\theta}}$, the two-faced risk on a data distribution $\mathcal{D}$ under the $\epsilon$-bounded perturbation is defined as follows:

$$\mathcal{R}_{\text{tf}}(f_{\boldsymbol{\theta}}, \mathcal{D}) = \mathbb{E}_{(\boldsymbol{x}, y) \sim \mathcal{D}} \left[ 1 - \min_{\|\boldsymbol{\delta}\| \leq \epsilon} \max_{\|\boldsymbol{t}\| \leq \epsilon} \mathbb{1}\left(f_{\boldsymbol{\theta}}(\boldsymbol{x} + \boldsymbol{\delta} + \boldsymbol{t}) \neq y\right) \right]. \quad (5)$$

In Eq. (5), it is intuitive to observe that for any instance $\boldsymbol{x}$ that is misclassified by $f_{\boldsymbol{\theta}}$, we have $1 - \min_{\|\boldsymbol{\delta}\| \leq \epsilon} \max_{\|\boldsymbol{t}\| \leq \epsilon} \mathbb{1}(f_{\boldsymbol{\theta}}(\boldsymbol{x} + \boldsymbol{\delta} + \boldsymbol{t}) \neq y) = 0$. This means that the two-faced risk of misclassified examples is always 0, and thus we should focus only on correctly classified examples. For the examples that are correctly classified by $f_{\boldsymbol{\theta}}$, we can further partition them into the examples that are correctly and robustly classified (denoted by $\mathcal{D}_{f_{\boldsymbol{\theta}}}^{\text{cr}}$) and the examples that are correctly classified but not robustly classified (denoted by $\mathcal{D}_{f_{\boldsymbol{\theta}}}^{\text{cnr}}$), where an example is considered correctly and robustly classified if it can still be correctly classified under an $\epsilon$-bounded perturbation. For the input $(\boldsymbol{x}, y)$ from distribution $\mathcal{D}_{f_{\boldsymbol{\theta}}}^{\text{cr}}$, we have $1 - \min_{\|\boldsymbol{\delta}\| \leq \epsilon} \max_{\|\boldsymbol{t}\| \leq \epsilon} \mathbb{1}(f_{\boldsymbol{\theta}}(\boldsymbol{x} + \boldsymbol{\delta} + \boldsymbol{t}) \neq y) = 1$. Therefore, we concentrate on the two-faced risk on $\mathcal{D}_{f_{\boldsymbol{\theta}}}^{\text{cnr}}$, i.e., $\mathcal{R}_{\text{tf}}(f_{\boldsymbol{\theta}}, \mathcal{D}_{f_{\boldsymbol{\theta}}}^{\text{cnr}})$.

Based on the above definition and analysis, we theoretically establish the relationship between two-faced risk and adversarial risk in the following theorem.

**Theorem 1.** $\mathcal{R}_{\text{tf}}(f_{\boldsymbol{\theta}}, \mathcal{D}) = 1 - \left(1 - \mathcal{R}_{\text{tf}}(f_{\boldsymbol{\theta}}, \mathcal{D}_{f_{\boldsymbol{\theta}}}^{\text{cnr}})\right) \cdot \mathcal{R}_{\text{adv}}(f_{\boldsymbol{\theta}}, \mathcal{D}) - \mathcal{R}_{\text{nat}}(f_{\boldsymbol{\theta}}, \mathcal{D}) \cdot \mathcal{R}_{\text{tf}}(f_{\boldsymbol{\theta}}, \mathcal{D}_{f_{\boldsymbol{\theta}}}^{\text{cnr}})$,

where $\mathcal{R}_{\text{nat}}(f_{\boldsymbol{\theta}}, \mathcal{D}) = \mathbb{E}_{(\boldsymbol{x}, y) \sim \mathcal{D}}[\mathbb{1}(f_{\boldsymbol{\theta}}(\boldsymbol{x}) \neq y)]$. The proof of Theorem 1 is provided in Appendix A. In Theorem 1, $\mathcal{R}_{\text{tf}}(f_{\boldsymbol{\theta}}, \mathcal{D}_{f_{\boldsymbol{\theta}}}^{\text{cnr}})$ can be considered as the success rate of two-faced attack, which provides a more meaningful measure of the resistance of the model to two-faced examples. To further empirically investigate the relationship between two-faced risk and adversarial risk, we construct models exhibiting varying levels of adversarial risk. This can be accomplished through the technique of adversarial training, which will be formally defined in the next section.

To investigate the association between two-faced risk and adversarial risk, we conducted two distinct experiments. *i)* We utilized TRADES with varying parameters $\lambda$ to generate ResNet-18 models with different adversarial risk on the CIFAR-10 and calculated the corresponding two-faced risk. The experimental results are depicted in Figure 2(a). *ii)* We employed different WideResNet-28-10 models from the RobustBench (Croce et al., 2021) and calculated their two-faced risk, respectively. The experimental results are depicted in Figure 2(b). Results show that two-faced risk and adversarial risk exhibit contrasting trends. Specifically, as the adversarial risk increases, the two-faced risk gradually decreases. This indicates an inherent difficulty in defending against the threat of two-faced attacks.

### 3.3 DISCUSSION ON COUNTERMEASURES

In this subsection, we discuss possible countermeasures to circumvent the threat of two-faced attacks. We have shown in Figure 1 that the models trained by conventional adversarial training methods are still vulnerable to two-faced examples. To mitigate the issue, here we present a simple adaptive defense by increasing the budget used in adversarial training.

Let us take PGD-AT (Madry et al., 2018) as an example. It works by leveraging projected gradient descent to generate adversarial examples and minimizing the cross-entropy loss on these examples:

$$\mathcal{L}_{\text{PGD-AT}} = \mathbb{E}_{(\boldsymbol{x}, y) \sim \mathcal{D}} \left[ \max_{\|\boldsymbol{\delta}\| \leq \epsilon} \ell_{\text{CE}} \left( f_{\boldsymbol{\theta}} \left( \boldsymbol{x} + \boldsymbol{\delta} \right), y \right) \right]. \tag{6}$$

Conventionally, the defense budget $\epsilon$ used in PGD-AT, and TRADES equals the budget used in evaluating adversarial robustness (Madry et al., 2018; Zhang et al., 2019). We propose to enlarge the defense budget $\epsilon$ to reduce the attack success rate of two-faced examples. Our intuition is that an $\epsilon$-bounded two-faced example can be successfully found due to the insufficiency of the conventional defense budget to ensure the model's decision boundary at an $\epsilon$-distance from unseen points. We postulate that a larger defense budget would be able to mitigate the issue.

Indeed, our experiments also validate the efficacy of this simple strategy in reducing two-faced risk. Specifically, we use PGD-AT to train models under $\ell_\infty$ norm with a larger defense budget ($12/255 \sim 20/255$). During the verification phase, we conduct two-faced attacks with $\epsilon = 8/255$. Results on CIFAR-10 are summarized in Figure 3. As a comparison, we also plot the performance of TRADES with larger regularization parameters, which has been shown to be effective in reducing two-faced risk in Figure 2. Appendix C provides detailed results. The default regularization parameter $\lambda$ in TRADES is set to 6 (Zhang et al., 2019). It turns out that the proposed adaptive defense achieves a better trade-off than TRADES with larger regularization parameters. In summary, countermeasures are effective in mitigating two-faced attacks, while the two-faced risk of models remains non-negligible.

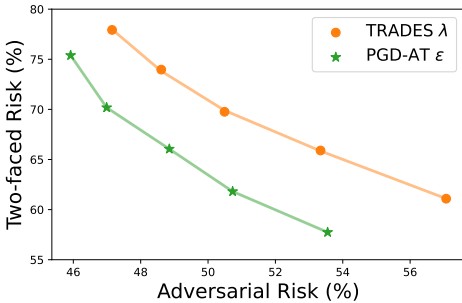

Figure 3: Comparison between countermeasures in terms of adversarial risk and two-faced risk under $\ell_\infty$ norm. Each point represents a model trained with different defense budgets $\epsilon$ or regularization parameters $\lambda$.

## 4 EXPERIMENTS

In this section, we conduct extensive experiments to demonstrate the effectiveness of two-faced attacks on different datasets and different network architectures. To demonstrate the threat of two-faced attacks more effectively, we assume that the adversary has access to the trained model and verification dataset to craft two-faced examples. The adversarial accuracy of the model on the verification dataset is used as the robustness metric of the model.

### 4.1 EXPERIMENTS ON ROBUSTLY TRAINED MODELS

We first train ResNet-18 models by employing three adversarial training methods, namely PGD-AT, TRADES, and THRM, across various datasets under $\ell_\infty$ threat model with $\epsilon = 8/255$ (or $\ell_2$ threat

Table 1: Robustness (%) of adversarially trained ResNet-18 under $\ell_\infty$ threat model.

| Dataset | Verification Examples | Poisoning (PGD-AT) | Poisoning (TRADES) | Poisoning (THRM) | Quality (PGD-AT) | Quality (TRADES) | Quality (THRM) |
|---------|----------------------|--------------------|--------------------|------------------|------------------|------------------|----------------|
| CIFAR-10 | Clean | 51.13 | 52.60 | 28.51 | 51.72 | 52.95 | 23.87 |
| | Hypocritical | 77.68 | 76.79 | 45.39 | 78.58 | 77.58 | 40.09 |
| | Two-faced | **81.36** | **79.50** | **77.67** | **82.81** | **80.19** | **75.54** |
| SVHN | Clean | 51.51 | 51.66 | 0.00 | 41.23 | 55.49 | 13.53 |
| | Hypocritical | 82.05 | 78.47 | 1.00 | 76.31 | 83.24 | 31.86 |
| | Two-faced | **86.70** | **82.91** | **4.51** | **79.99** | **86.70** | **56.46** |
| CIFAR-100 | Clean | 28.23 | 29.53 | 1.13 | 28.85 | 29.98 | 19.04 |
| | Hypocritical | 52.08 | 53.02 | **10.85** | 52.30 | 53.54 | 42.31 |
| | Two-faced | **56.08** | **55.87** | 9.86 | **55.31** | **56.41** | **56.52** |
| Tiny-ImageNet | Clean | 21.76 | 21.12 | 20.55 | 21.99 | 21.77 | 19.28 |
| | Hypocritical | 42.19 | 44.10 | 41.86 | 44.12 | 45.30 | 41.39 |
| | Two-faced | **44.28** | **45.65** | **44.67** | **46.72** | **47.18** | **46.37** |

Table 2: Robustness (%) of adversarially trained models under $\ell_\infty$ threat model on CIFAR-10.

| Verification Examples | Carmon et al. (2019) | Xu et al. (2023) | Debenedetti et al. (2022) | Dai et al. (2021) |
|-----------------------|----------------------|------------------|---------------------------|-------------------|
| Clean | 62.60 | 68.59 | 59.71 | 64.40 |
| Hypocritical | 84.65 | 77.93 | 83.77 | 84.63 |
| Two-faced | **89.14** | **88.70** | **90.16** | **86.63** |
| | Rice et al. (2020) | Wong et al. (2020) | Zhang et al. (2019) | Zhang et al. (2020) |
| Clean | 57.41 | 46.87 | 55.35 | 57.30 |
| Hypocritical | 81.98 | 76.89 | 80.37 | 80.50 |
| Two-faced | **85.00** | **81.71** | **84.16** | **83.74** |

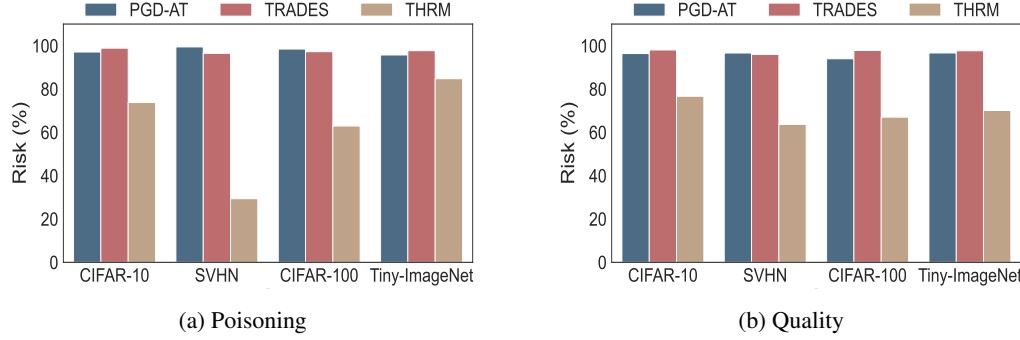

(a) Poisoning  (b) Quality

Figure 4: The attack success rate of two-faced attack of the adversarially trained models.

model with $\epsilon = 0.5$). The number of PGD iterations is set to 10 and 20 for training and verification time, respectively. 'Quality' represents the original training dataset, and 'Poisoning' represents the training dataset that has been perturbed to maximize adversarial risk under $\ell_\infty$ threat model with $\epsilon = 8/255$ (or $\ell_2$ threat model with $\epsilon = 0.5$), and the number of iterations for PGD is set to 100 (Tao et al., 2021). More experimental details are provided in Appendix B.

Table 1 shows the robustness of models with $\ell_\infty$ threat model on different types of verification examples (i.e., clean examples, hypocritical examples, and two-faced examples). The "Clean" row represents the real robustness of models. It turns out that all three adversarial training methods obtain moderate robustness against adversarial perturbations. The adversarially trained models exhibit a noticeable improvement in adversarial robustness on hypocritical and two-faced examples. The models exhibit their highest level of robustness on two-faced examples, because hypocritical attacks focus on the accuracy of models, while two-faced attacks focus on the robustness of models. For instance, on CIFAR-10, the Quality (PGD-AT) model shows a robustness (adversarial accuracy) of

Table 3: Robustness (%) of adversarially trained ResNet-18 uder $\ell_2$ threat model.

| Dataset | Verification Examples | Poisoning (PGD-AT) | Poisoning (TRADES) | Quality (PGD-AT) | Quality (TRADES) |
|---|---|---|---|---|---|
| CIFAR-10 | Clean | 69.04 | 70.07 | 69.85 | 70.98 |
| | Hypocritical | 87.06 | 85.12 | 87.39 | 85.90 |
| | Two-faced | **88.92** | **86.01** | **89.26** | **86.89** |
| SVHN | Clean | 67.57 | 69.51 | 67.56 | 69.86 |
| | Hypocritical | 90.63 | 88.79 | 90.95 | 89.66 |
| | Two-faced | **93.44** | **91.57** | **94.19** | **92.69** |
| CIFAR-100 | Clean | 41.28 | 43.06 | 41.34 | 43.96 |
| | Hypocritical | 61.89 | 60.07 | 62.51 | 61.14 |
| | Two-faced | **64.85** | **61.25** | **65.42** | **62.32** |
| Tiny-ImageNet | Clean | 44.78 | 44.90 | 60.01 | 45.83 |
| | Hypocritical | 59.13 | 57.87 | **74.18** | 58.60 |
| | Two-faced | **60.18** | **58.29** | 73.79 | **58.94** |

Table 4: Robustness (%) of adversarially trained ResNet-18 under different verification methods.

| Training Method | Threat Model | Verification Examples | DI-FGSM | APGD$_{CE}$ | FAB$^{T}$ | Square |
|---|---|---|---|---|---|---|
| PGD-AT | $\ell_\infty$ | Clean | 55.01 | 51.00 | 48.60 | 55.02 |
| | | Two-faced | **84.79** | **81.47** | **77.39** | **82.68** |
| | $\ell_2$ | Clean | - | 69.54 | 69.27 | 80.09 |
| | | Two-faced | - | **88.38** | **88.04** | **93.20** |
| TRADES | $\ell_\infty$ | Clean | 54.71 | 52.43 | 49.89 | 53.81 |
| | | Two-faced | **81.40** | **79.39** | **76.31** | **79.39** |
| | $\ell_2$ | Clean | - | 70.81 | 69.89 | 78.29 |
| | | Two-faced | - | **86.63** | **85.78** | **90.65** |

78.58% on hypocritical examples and 82.81% on two-faced examples, whereas its real robustness stands at a lower rate of 51.72%. Table 3 show the robustness of models with $\ell_2$ threat model, and similar conclusions can be drawn. Additionally, we evaluated several robust models sourced from the publicly accessible RobustBench. The results are shown in Table 2. As can be observed from Table 2 that even for models with different architectures or adversarial training methods, they exhibit higher false robustness on two-faced examples. Resultes in Appendix C also indicate that the elevated false robustness achieved by two-faced attacks does not come at the expense of reduced accuracy; on the contrary, it even leads to falsely elevated accuracy.

In order to compare the threat of two-faced attacks against different adversarially trained models intuitively, we illustrate the attack success rate of two-faced attacks in Figure 4. As can be seen from Figure 4, THRM has a lower attack success rate than PGD-AT and TRADES, which suggests that THRM may exhibit a certain level of resilience against two-faced attacks. Another intriguing observation is that as the adversarial risk of the model increases, the attack success rate of the two-faced attacks diminishes. For example, it can be seen from Table 1 that the adversarial risk of PGD-AT is higher than that of TRADES on CIFAR-10 (i.e., PGD-AT achieves lower robust accuracy on clean examples than TRADES), while in Figure 4(a), the two-faced attack success rate of PGD-AT is higher than that of TRADES. This phenomenon is consistent with our analysis in Section 3.2: models that have lower adversarial risks tend to have higher two-faced risks.

Furthermore, we are curious about whether the two-faced attack is effective against different robustness verification methods. We first generate two-faced examples for adversarially trained models and then use three white-box verification methods: DI-FGSM (Xie et al., 2018), APGD$_{CE}$ (Croce & Hein, 2020), and FAB$^{T}$ (Croce & Hein, 2020), as well as one black-box verification method: Square (Andriushchenko et al., 2020), implemented within the torchattacks library (Kim, 2020), on both Clean and Two-faced examples (keeping the perturbation norm and budget consistent with the training methods). Table 4 presents the relevant results on CIFAR-10. It can be observed from Table 4 that the same two-faced examples, under different robustness verification methods, consis-

Table 5: Transferability of Two-faced Attacks on CIFAR-10 under $\ell_\infty$ threat model.

|  | Clean | MLP | VGG-16 | ResNet-18 | WideResNet-28-10 |
|---|---|---|---|---|---|
| MLP | 27.51 | **49.9** | 32.82 | 32.04 | 30.85 |
| VGG-16 | 48.51 | 56.32 | **79.28** | 69.77 | 65.02 |
| ResNet-18 | 51.72 | 58.18 | 73.55 | **84.79** | 71.62 |
| WideResNet-28-10 | 53.91 | 58.4 | 72.06 | 74.89 | **85.93** |

Table 6: Robustness (%) of naturally trained MLP and ResNet-18 under $\ell_2$ threat model.

| Dataset | Verification Examples | MLP | | ResNet-18 | |
|---|---|---|---|---|---|
|  |  | Poisoning | Quality | Poisoning | Quality |
| CIFAR-10 | Clean | 0.73 | 19.68 | 0.00 | 0.08 |
|  | Hypocritical | 24.93 | 54.23 | 0.16 | 0.91 |
|  | Two-faced | **31.00** | **57.54** | **5.19** | **17.38** |
| SVHN | Clean | 4.52 | 30.07 | 0.00 | 9.43 |
|  | Hypocritical | 47.73 | 73.04 | 0.01 | 14.90 |
|  | Two-faced | **52.98** | **76.71** | **0.51** | **53.00** |
| CIFAR-100 | Clean | 4.44 | 12.07 | 0.00 | 0.18 |
|  | Hypocritical | 20.45 | 31.07 | **0.16** | 2.44 |
|  | Two-faced | **25.92** | **32.83** | 0.12 | **7.47** |
| Tiny-ImageNet | Clean | 6.28 | 6.45 | 0.00 | 2.92 |
|  | Hypocritical | 13.50 | 13.95 | **1.95** | 19.18 |
|  | Two-faced | **13.60** | **14.16** | 0.08 | **37.27** |

tently exhibit higher robustness. This indicates that the two-faced attack is capable of generating falsely elevated verification robustness across different robustness verification methods.

Table 5 shows the transferability of two-faced attacks. All models in the table are trained with PGD-AT ($\ell_\infty$). We employ PGD ($\ell_\infty$) as the robustness verification method. The results for $\ell_2$ are shown in Appendix C. Cell $(i, j)$ indicates the verification robustness of model $i$ on two-faced examples generated by model $j$. It can be observed that within each row, model $i$ exhibits the highest verification robustness on two-faced examples generated by itself. Additionally, the verification robustness on two-faced examples generated by various models is consistently higher than the real robustness. This indicates that the two-faced attacks possess a certain degree of transferability.

## 4.2 Experiments on Naturally Trained Models

The above experimental results show that two-faced attacks are a great threat to adversarially trained models. Here, we are curious about whether two-faced attacks are still effective for naturally trained models to trick researchers into deploying substandard models. We naturally train MLP and ResNet-18 on Poisoning and Quality datasets. More experimental details are provided in Appendix D.

Table 6 shows the robustness of naturally trained MLP and ResNet-18 on various datasets. The table indicates that both models show improved verification robustness on hypocritical and two-faced examples compared to Clean examples, with two-faced examples being the highest in most case. It's worth noting that both models exhibit low robustness on Clean examples, especially the model trained on Poisoning data, where its robustness approaches zero. This is because naturally trained models rely on non-robust features to improve accuracy, resulting in lower robustness (Tsipras et al., 2019). Poisoning data is intentionally designed to reduce the robustness of models trained on it, hence these models have near-zero robustness on clean verification examples. However, in contrast to adversarially trained models, naturally trained models exhibit very low false robustness on two-faced verification examples. For instance, the false robustness of naturally trained ResNet-18 on two-faced verification examples is even lower than the true robustness of adversarially trained ResNet-18 on Clean examples. This suggests that deceiving researchers into deploying these low-

robustness models might be challenging. Results of naturally trained models under $\ell_\infty$ threat model on different datasets and naturally trained different architectures on CIFAR-10 are provided in Appendix D, and similar conclusions hold. Visualizations of perturbations in Appendix D show that two-faced attacks enhance the robust features of verification examples for adversarially trained models. However, for naturally trained models, the perturbations lack distinct robust features. This can explain why two-faced attacks pose a less threat to naturally trained models.

## 5 RELATED WORK

In this section, we review related studies on relevant topics, including adversarial examples, hypocritical examples, and adversarial training.

**Adversarial Examples.** Adversarial examples refer to the examples formed by intentionally introducing subtle perturbations into the dataset, causing the model to output an incorrect prediction with high confidence (Szegedy et al., 2014). The threat posed by adversarial examples lies in their ability to easily deceive deep learning models, even those trained on large-scale data with high accuracy (Nguyen et al., 2014). For instance, in the field of image recognition, making minor pixel modifications to the input image can lead the model to misclassify it as a different category (Moosavi-Dezfooli et al., 2016a). Since adversarial examples can pose a significant challenge to security, many attack techniques have been designed to find them (Goodfellow et al., 2015; Papernot et al., 2016; Moosavi-Dezfooli et al., 2016b; Kurakin et al., 2016; Dong et al., 2018; Chen et al., 2018; Wang et al., 2020a; Croce & Hein, 2020). Most of the previous works focus on reducing the performance of well-trained models, while this paper aims to hypocritically improve the verification robustness of adversarially trained models.

**Hypocritical Examples.** A recent study introduced a new type of threat called hypocritical examples (Tao et al., 2022b), which refers to initially misclassified inputs that are subtly manipulated by a false friend, causing the model to correctly predict the labels. Hypocritical examples cannot be simply ignored before model deployment because they can be maliciously exploited during the evaluation process to conceal the errors of inaccurate models. If deployers trust the performance of the evaluated model on hypocritical examples and utilize such seemingly high-performing models in real-world scenarios, unexpected failures may be encountered. Later work further found that hypocritical examples can be used as poisons in the training process to produce poorly-performed models (Tao et al., 2021; Huang et al., 2021; Fu et al., 2022; Tao et al., 2022a). It is noteworthy that hypocritical examples focus on the natural accuracy of models, while our studied two-faced examples focus on the adversarial accuracy (robustness) of models.

**Adversarial Training.** Adversarial training is a highly effective approach for enhancing the robustness of deep learning models. It involves the generation of adversarial examples using specialized attack methods during the training phase. These adversarial examples are then combined with normal examples to update the model parameters, enabling the model to adapt and defend against such perturbations. Previous studies have introduced various adversarial training methods (Goodfellow et al., 2015; Madry et al., 2018; Zhang et al., 2019; Wang et al., 2020b; Zhang et al., 2020; Wu et al., 2020; Zhang et al., 2020; Pang et al., 2021). Although these adversarial training methods can effectively resist the influence of adversarial examples, they cannot fully resist the influence of the two-faced examples proposed in this paper.

## 6 CONCLUSION

In this paper, we for the first time showed that adversarially trained models are vulnerable to a new threat in the verification phase called two-faced attacks, where slightly perturbed features of input data can make the model exhibit a false sense of adversarial robustness. Such a threat is significantly important as it can mislead our evaluation of the adversarial robustness of models, which could cause unpredictable security issues when deploying substandard models in reality. More seriously, we found that many types of substandard models suffer from this threat, which means this threat may be pervasive. We provided formulations of two-faced examples and two-faced risk. We theoretically and empirically showed that two-faced risk and adversarial risk exhibit contrasting trends. We also tried to circumvent it via a simple countermeasure by enlarging the budget used in adversarial training. Experimental results consistently supported our claims.

ACKNOWLEDGMENTS

Tao Xiang is supported by the National Key R&D Program of China under Grant 2022YFB3103500. Bo An is supported by the National Research Foundation, Singapore under its Industry Alignment Fund – Pre-positioning (IAF-PP) Funding Initiative. Any opinions, findings and conclusions or recommendations expressed in this material are those of the author(s) and do not reflect the views of National Research Foundation, Singapore. Lei Feng is supported by the Chongqing Overseas Chinese Entrepreneurship and Innovation Support Program.

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

## A    PROOF OF THEOREM 1

**Theorem 2.** (**restated**) $\mathcal{R}_{\mathrm{tf}}\left(f_{\boldsymbol{\theta}}, \mathcal{D}\right) = 1 - \left(1 - \mathcal{R}_{\mathrm{tf}}\left(f_{\boldsymbol{\theta}}, \mathcal{D}_{f_{\boldsymbol{\theta}}}^{\mathrm{cnr}}\right)\right) \cdot \mathcal{R}_{\mathrm{adv}}\left(f_{\boldsymbol{\theta}}, \mathcal{D}\right) - \mathcal{R}_{\mathrm{nat}}\left(f_{\boldsymbol{\theta}}, \mathcal{D}\right) \cdot \mathcal{R}_{\mathrm{tf}}\left(f_{\boldsymbol{\theta}}, \mathcal{D}_{f_{\boldsymbol{\theta}}}^{\mathrm{cnr}}\right)$.

*Proof.* To simplify the presentation, we let $f$ denote $f_{\boldsymbol{\theta}}$ and $\mathbb{1}\left(f\left(\boldsymbol{x}''\right) = y\right) = \max_{\|\boldsymbol{\delta}\| \leq \epsilon} \min_{\|\boldsymbol{t}\| \leq \epsilon} \mathbb{1}\left(f\left(\boldsymbol{x} + \boldsymbol{\delta} + \boldsymbol{t}\right) = y\right)$. We have:

$$\mathcal{R}_{\mathrm{tf}}\left(f, \mathcal{D}\right)$$

$$= \mathbb{E}_{(\boldsymbol{x},y)\sim\mathcal{D}}\left[1 - \min_{\|\boldsymbol{\delta}\| \leq \epsilon} \max_{\|\boldsymbol{t}\| \leq \epsilon} \mathbb{1}\left(f\left(\boldsymbol{x} + \boldsymbol{\delta} + \boldsymbol{t}\right) \neq y\right)\right]$$

$$= \mathbb{E}_{(\boldsymbol{x},y)\sim\mathcal{D}}\left[\max_{\|\boldsymbol{\delta}\| \leq \epsilon} \min_{\|\boldsymbol{t}\| \leq \epsilon} \mathbb{1}\left(f\left(\boldsymbol{x} + \boldsymbol{\delta} + \boldsymbol{t}\right) = y\right)\right]$$

$$= \mathbb{E}_{(\boldsymbol{x},y)\sim\mathcal{D}}\left[\mathbb{1}\left(f\left(\boldsymbol{x}''\right) = y\right)\right]$$

$$= \mathbb{E}_{(\boldsymbol{x},y)\sim\mathcal{D}}\left[\mathbb{1}\left(f\left(\boldsymbol{x}''\right) = y\right) \cdot 1\right]$$

$$= \mathbb{E}_{(\boldsymbol{x},y)\sim\mathcal{D}}\left[\mathbb{1}\left(f\left(\boldsymbol{x}''\right) = y\right) \cdot \left(\mathbb{1}\left(f\left(\boldsymbol{x}_{\mathrm{adv}}\right) = y\right) + \mathbb{1}\left(f\left(\boldsymbol{x}_{\mathrm{adv}}\right) \neq y\right)\right)\right]$$

$$= \mathbb{E}_{(\boldsymbol{x},y)\sim\mathcal{D}}\left[\mathbb{1}\left(f\left(\boldsymbol{x}''\right) = y\right) \cdot \mathbb{1}\left(f\left(\boldsymbol{x}_{\mathrm{adv}}\right) = y\right)\right] + \mathbb{E}_{(\boldsymbol{x},y)\sim\mathcal{D}}\left[\mathbb{1}\left(f\left(\boldsymbol{x}''\right) = y\right) \cdot \mathbb{1}\left(f\left(\boldsymbol{x}_{\mathrm{adv}}\right) \neq y\right)\right]$$

$$= \mathbb{E}_{(\boldsymbol{x},y)\sim\mathcal{D}}\left[\mathbb{1}\left(f\left(\boldsymbol{x}_{\mathrm{adv}}\right) = y\right)\right] + \mathbb{E}_{(\boldsymbol{x},y)\sim\mathcal{D}}\left[\mathbb{1}\left(f\left(\boldsymbol{x}_{\mathrm{adv}}\right) \neq y\right)\right] \cdot \mathbb{E}_{(\boldsymbol{x},y)\sim\mathcal{D}_f^{\mathrm{cnr}}}\left[\mathbb{1}\left(f\left(\boldsymbol{x}''\right) = y\right)\right]$$

$$\qquad - \mathbb{E}_{(\boldsymbol{x},y)\sim\mathcal{D}}\left[\mathbb{1}\left(f\left(\boldsymbol{x}\right) \neq y\right)\right] \cdot \mathbb{E}_{(\boldsymbol{x},y)\sim\mathcal{D}_f^{\mathrm{cnr}}}\left[\mathbb{1}\left(f\left(\boldsymbol{x}''\right) = y\right)\right]$$

$$= 1 - \mathbb{E}_{(\boldsymbol{x},y)\sim\mathcal{D}}\left[\mathbb{1}\left(f\left(\boldsymbol{x}_{\mathrm{adv}}\right) \neq y\right)\right] + \mathbb{E}_{(\boldsymbol{x},y)\sim\mathcal{D}}\left[\mathbb{1}\left(f\left(\boldsymbol{x}_{\mathrm{adv}}\right) \neq y\right)\right] \cdot \mathbb{E}_{(\boldsymbol{x},y)\sim\mathcal{D}_f^{\mathrm{cnr}}}\left[\mathbb{1}\left(f\left(\boldsymbol{x}''\right) = y\right)\right]$$

$$\qquad - \mathbb{E}_{(\boldsymbol{x},y)\sim\mathcal{D}}\left[\mathbb{1}\left(f\left(\boldsymbol{x}\right) \neq y\right)\right] \cdot \mathbb{E}_{(\boldsymbol{x},y)\sim\mathcal{D}_f^{\mathrm{cnr}}}\left[\mathbb{1}\left(f\left(\boldsymbol{x}''\right) = y\right)\right]$$

$$= 1 - \mathcal{R}_{\mathrm{adv}}\left(f, \mathcal{D}\right) + \mathcal{R}_{\mathrm{adv}}\left(f, \mathcal{D}\right) \cdot \mathcal{R}_{\mathrm{tf}}\left(f, \mathcal{D}_f^{\mathrm{cnr}}\right) - \mathcal{R}_{\mathrm{nat}}\left(f, \mathcal{D}\right) \cdot \mathcal{R}_{\mathrm{tf}}\left(f, \mathcal{D}_f^{\mathrm{cnr}}\right)$$

$$= 1 - \left(1 - \mathcal{R}_{\mathrm{tf}}\left(f, \mathcal{D}_f^{\mathrm{cnr}}\right)\right) \cdot \mathcal{R}_{\mathrm{adv}}\left(f, \mathcal{D}\right) - \mathcal{R}_{\mathrm{nat}}\left(f, \mathcal{D}\right) \cdot \mathcal{R}_{\mathrm{tf}}\left(f, \mathcal{D}_f^{\mathrm{cnr}}\right)$$

$\square$

## B    EXPERIMENTAL DETAILS

We conduct all experiments using NVIDIA GeForce RTX 3090 GPUs. We implement the experimental code using PyTorch.

### B.1    TRAINING DETAILS

**CIFAR-10.** CIFAR-10 (Krizhevsky, 2009) is a widely used image dataset in computer vision research. It consists of 60,000 $32 \times 32$ pixel color images (50,000 images for training and 10,000 images for testing), representing 10 different classes. We take the original training data as the Qualify training data and use the method in Appendix B.4 to generate the Poisoning training data. We use the original test data set as the Clean validation data. For the training set, we use $32 \times 32$ random cropping with a 4-pixel padding and random horizontal flip as the data augmentation method. We train ResNet-18 (He et al., 2015), DenseNet-121 (Huang et al., 2016), and WideResNet-28-10 (Zagoruyko & Komodakis, 2016) models using SGD with a learning rate of 0.1, momentum of 0.9, and weight decay of $5 \times 10^{-4}$. Additionally, the MLP and VGG-16 (Simonyan & Zisserman, 2014) models are trained with SGD using a learning rate of 0.01, momentum of 0.9, and weight decay of $5 \times 10^{-4}$. For all architectures, the training epoch is fixed at 110 with batch size 128 and learning rate was decayed by a factor of 0.1 in the 100-th epoch and the 105-th epoch respectively.

**CIFAR-100.** CIFAR-100 (Krizhevsky, 2009) is a widely used image dataset in computer vision research. It consists of 60,000 $32 \times 32$ pixel color images (50,000 images for training and 10,000

images for testing), representing 100 different classes. We take the original training data as the Qualify training data and use the method in Appendix B.4 to generate the Poisoning training data. We use the original test data set as the Clean validation data. For the training set, we use $32 \times 32$ random cropping with a 4-pixel padding and random horizontal flip as the data augmentation method. We train ResNet-18 models using SGD with a learning rate of 0.1, momentum of 0.9, and weight decay of $5 \times 10^{-4}$. Additionally, the MLP models are trained with SGD using a learning rate of 0.01, momentum of 0.9, and weight decay of $5 \times 10^{-4}$. For all architectures, the training epoch is fixed at 110 with batch size 128 and learning rate was decayed by a factor of 0.1 in the 100-th epoch and the 105-th epoch respectively.

**SVHN.** SVHN (Netzer et al., 2011) is a widely used image dataset in computer vision research. It consists of 630,420 $32 \times 32$ pixel color images (73,257 images for training and 26,032 images for testing), representing 10 different classes. We take the original training data as the Qualify training data and use the method in Appendix B.4 to generate the Poisoning training data. We use the original test data set as the Clean validation data. For the training set, we use $32 \times 32$ random cropping with a 4-pixel padding and random horizontal flip as the data augmentation method. We train ResNet-18 and MLP models using SGD with a learning rate of 0.01, momentum of 0.9, and weight decay of $5 \times 10^{-4}$. For all architectures, the training epoch is fixed at 60 with batch size 128 and learning rate was decayed by a factor of 0.1 in the 50-th epoch and the 55-th epoch respectively.

**Tiny-ImageNet.** Tiny-ImageNet (Yao et al., 2015) is a widely used image dataset in computer vision research. It consists of 110000 $64 \times 64$ pixel color images (100,000 images for training and 100,000 images for testing), representing 200 different classes. We take the original training data as the Qualify training data and use the method in Appendix B.4 to generate the Poisoning training data. We use the original test data set as the Clean validation data. For the training set, we use $64 \times 64$ random cropping with a 4-pixel padding and random horizontal flip as the data augmentation method. We train ResNet-18 and MLP models using SGD with a learning rate of 0.1, momentum of 0.9, and weight decay of $5 \times 10^{-4}$. For all architectures, the training epoch is fixed at 60 with batch size 64 and learning rate was decayed by a factor of 0.1 in the 50-th epoch and the 55-th epoch respectively.

## B.2   ADVERSARIAL TRAINING

We perform robust training algorithms including PGD-AT, TRADES, and THRM by following the common settings (Madry et al., 2018; Pang et al., 2021). Specifically, we use projected gradient descent (PGD) with random initial perturbations. We set the PGD step to 10 and the step size to $\epsilon/4$. We employ the $\ell_2$ and $\ell_\infty$ norms for $\ell_2$ with $\epsilon = 0.5$ and for $\ell_\infty$ with $\epsilon = 8/255$. Additionally, unless otherwise specified, the default value for the $\lambda$ parameter in TRADES and THRM is set to 5.

## B.3   VERIFICATION DETAILS

During the verification of two-faced attacks and adversarial attacks, we set the PGD step to 20 and the step size to $\epsilon/4$. We employ the $\ell_2$ and $\ell_\infty$ norms for $\ell_2$ with $\epsilon = 0.5$ and for $\ell_\infty$ with $\epsilon = 8/255$.

## B.4   POISONING DATASET

The creation of the Poisoning dataset involves modifying images to increase generalization error while preserving the original labels (Ilyas et al., 2019). Each $(\boldsymbol{x}, y)$ pair in the Poisoning dataset is generated as follows: a target class $t$ is deterministically chosen based on the source class $y$, using a predetermined permutation of labels. Subsequently, a slight adversarial perturbation is applied to $\boldsymbol{x}$, aiming to cause its misclassification as $t$ by a naturally trained model. The perturbations are constrained within the $\ell_2$-norm with $\epsilon = 0.5$ or $\ell_\infty$-norm with $\epsilon = 8/255$. The PGD is employed with 100 iterations and a step size of $\epsilon/4$.

Table 7: Experiments of models with different robustness on CIFAR-10. Models were trained on Quality data by modifying the TRADES parameter $\lambda$ under the $\ell_\infty$ threat model.

| $\lambda$ | Robustness | | | Risk | | |
|---|---|---|---|---|---|---|
| | Clean | Hypocritical | Two-faced | $R_{\text{nat}}(f_{\boldsymbol{\theta}}, D)$ | $R_{\text{adv}}(f_{\boldsymbol{\theta}}, D_{f_{\boldsymbol{\theta}}}^+)$ | $R_{\text{tf}}(f_{\boldsymbol{\theta}}, D_{f_{\boldsymbol{\theta}}}^{\text{cnr}})$ |
| 0 | 0.0 | 0.0 | 0.02 | 6.08 | 100.00 | 0.02 |
| 2 | 50.97 | 80.95 | 84.79 | 14.16 | 40.59 | 96.98 |
| 3 | 52.42 | 80.0 | 83.61 | 15.64 | 37.78 | 97.64 |
| 4 | 52.66 | 79.28 | 82.24 | 17.08 | 36.46 | 97.75 |
| 5 | 52.92 | 78.38 | 81.11 | 18.21 | 35.31 | 97.64 |
| 10 | 52.82 | 75.6 | 77.93 | 21.64 | 32.66 | 98.31 |
| 20 | 51.39 | 72.02 | 73.97 | 25.68 | 30.93 | 98.47 |
| 40 | 49.51 | 68.26 | 69.76 | 29.84 | 29.35 | 98.05 |
| 60 | 46.66 | 64.45 | 65.9 | 33.61 | 29.75 | 97.52 |
| 80 | 42.93 | 59.8 | 61.09 | 38.66 | 29.96 | 98.64 |
| 100 | 46.34 | 62.97 | 64.18 | 35.42 | 28.30 | 97.81 |

Table 8: Transferability of Two-faced Attacks on CIFAR-10 with $\ell_2$ perturbation. All models in the table are trained using PGD-AT ($\ell_2$). We employ PGD ($\ell_2$) as the robustness verification method.

| | Clean | MLP | VGG-16 | ResNet-18 | WideResNet-28-10 |
|---|---|---|---|---|---|
| MLP | 24.64 | **58.17** | 27.87 | 27.61 | 26.65 |
| VGG-16 | 66.08 | 69.59 | **87.34** | 80.05 | 77.95 |
| ResNet-18 | 69.85 | 70.4 | 82.54 | **89.26** | 82.09 |
| WideResNet-28-10 | 70.8 | 73.13 | 82.2 | 83.63 | **90.91** |

# C EXPERIMENTS ON ROBUSTLY TRAINED MODELS

## C.1 OMITTED TABLES

## C.2 VISUALIZATION OF PERTURBATION

In order to gain a deeper understanding of two-faced attacks, we visualize two perturbations for both the non-robust and robust models. Figure 5(a) presents the results of the non-robust model obtained from applying natural training (NT) on SVHN. Figure 5(b) presents the results of the robust model obtained from employing PDG-AT (under $\ell_\infty$ threat model with $\epsilon = 8/255$) on SVHN. Figure 5(c) presents the results of the robust model obtained from employing TRADES (under $\ell_\infty$ threat model with $\epsilon = 8/255$ and $\lambda = 6$) on SVHN. We observed that both hypocritical and two-faced attacks produce chaotic perturbations on the non-robust model in Figure 5(a), while the two attacks in Figure 5(b) and Figure 5(c) could see the contour information of the original picture when the perturbations were generated on the two robust models. Since the generation of perturbations requires the calculation of the gradient of the input image, we believe that the gradient information of the models generated by adversarial training contains some effective information about the original data, while the information of naturally trained models is little. This phenomenon could explain why two-faced attacks are effective for adversarially trained models but not for naturally trained models.

## C.3 ACCURACY OF ROBUSTLY TRAINED MODELS

Table 9 to 11 show the accuracy of adversarially trained models on Clean verification examples and two-faced examples. From these tables, it can be observed that the accuracy on two-faced verification examples is consistently higher than that on Clean verification examples. This indicates that our proposed two-faced attack not only introduces false higher robustness to adversarially trained models during verification but also results in false higher accuracy.

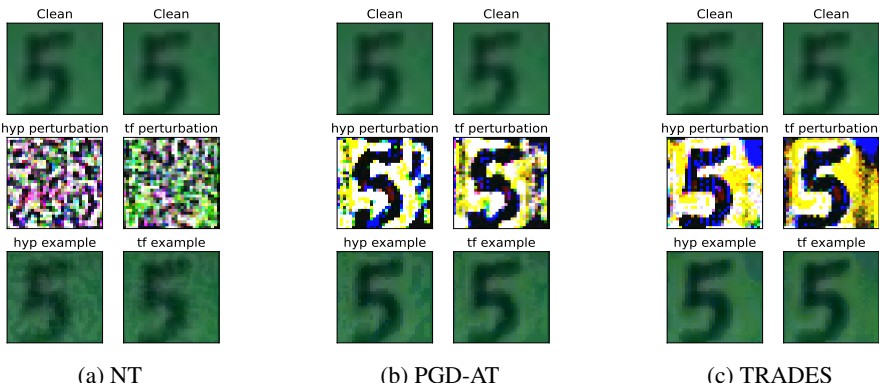

(a) NT    (b) PGD-AT    (c) TRADES

Figure 5: Visualization of the clean examples, hypocritical examples, and two-faced examples on the SVHN dataset. (a) Results of the natural trained (NT) model; (b) Results of the model trained with PGD-AT; (c) Results of the model trained with TRADES. The perturbations in (a) are chaotic, while those in (b) and (c) exhibit clear semantics.

Table 9: Accuracy (%) of adversarially trained (PGD-AT) models under different threat model on CIFAR-10.

| Threat Model | Verification Examples | MLP | VGG-16 | ResNet-18 | DenseNet-121 | WideResNet-28-10 |
|---|---|---|---|---|---|---|
| $\ell_\infty$ | Clean | 50.33 | 80.02 | 83.97 | 81.74 | 87.84 |
| | Two-faced | **71.36** | **93.83** | **95.89** | **94.56** | **97.65** |
| $\ell_2$ | Clean | 58.31 | 92.94 | 89.39 | 88.85 | 95.81 |
| | Two-faced | **64.11** | **96.83** | **96.85** | **96.47** | **98.09** |

Table 10: Accuracy (%) of adversarially trained ResNet-18 under $\ell_\infty$ threat model.

| Dataset | Verification Examples | Poisoning (PGD-AT) | Poisoning (TRADES) | Quality (PGD-AT) | Quality (TRADES) |
|---|---|---|---|---|---|
| CIFAR-10 | Clean | 82.26 | 79.8 | 83.97 | 80.73 |
| | Two-faced | **95.07** | **94.00** | **95.89** | **94.41** |
| SVHN | Clean | 86.89 | 84.05 | 81.33 | 88.01 |
| | Two-faced | **97.99** | **97.01** | **96.90** | **97.98** |
| CIFAR-100 | Clean | 57.79 | 56.61 | 56.51 | 57.00 |
| | Two-faced | **80.41** | **79.82** | **77.36** | **80.54** |
| Tiny-ImageNet | Clean | 45.28 | 46.22 | 47.57 | 47.78 |
| | Two-faced | **66.41** | **71.20** | **68.63** | **72.90** |

Table 11: Accuracy (%) of adversarially trained ResNet-18 with $\ell_2$ perturbation.

| Dataset | Verification Examples | Poisoning (PGD-AT) | Poisoning (TRADES) | Quality (PGD-AT) | Quality (TRADES) |
|---|---|---|---|---|---|
| CIFAR-10 | Clean | 89.06 | 85.95 | 89.39 | 86.84 |
| | Two-faced | **96.59** | **94.10** | **96.85** | **94.74** |
| SVHN | Clean | 93.71 | 91.67 | 94.51 | 92.78 |
| | Two-faced | **98.72** | **97.99** | **98.96** | **98.35** |
| CIFAR-100 | Clean | 65.07 | 61.25 | 65.71 | 62.35 |
| | Two-faced | **82.70** | **77.20** | **83.20** | **77.96** |
| Tiny-ImageNet | Clean | 60.20 | 58.36 | 60.96 | 59.07 |
| | Two-faced | **73.22** | **69.89** | **74.46** | **71.30** |

Table 12: Experiments of ResNet-18 with different robustness (%) on CIFAR-10. Models were trained on Quality data by modifying the THRM parameter $\lambda$ under the $\ell_\infty$ threat model.

| $\lambda$ | Robustness | | Risk | | |
|---|---|---|---|---|---|
| | Clean | Two-faced | $R_{\mathrm{nat}}(f_{\boldsymbol{\theta}}, D)$ | $R_{\mathrm{adv}}(f_{\boldsymbol{\theta}}, D^+_{f_{\boldsymbol{\theta}}})$ | $R_{\mathrm{tf}}(f_{\boldsymbol{\theta}}, D^{\mathrm{cnr}}_{f_{\boldsymbol{\theta}}})$ |
| 1 | 4.53 | 39.55 | 5.56 | 95.20 | 38.95 |
| 5 | 23.87 | 75.54 | 8.75 | 73.84 | 76.68 |
| 10 | 32.75 | 79.5 | 11.83 | 62.85 | 84.35 |
| 20 | 35.85 | 73.14 | 9.79 | 60.25 | 68.59 |
| 40 | 43.22 | 76.03 | 20.59 | 45.57 | 90.66 |
| 60 | 43.66 | 73.02 | 24.56 | 42.12 | 92.38 |
| 80 | 41.92 | 71.98 | 25.22 | 43.94 | 91.47 |
| 100 | 43.9 | 69.51 | 29.08 | 38.09 | 94.78 |

Table 13: Experiments of ResNet-18 with different robustness (%) on CIFAR-100. Models were trained on Quality data by modifying the THRM parameter $\lambda$ under the $\ell_\infty$ threat model.

| $\lambda$ | Robustness | | Risk | | |
|---|---|---|---|---|---|
| | Clean | Two-faced | $R_{\mathrm{nat}}(f_{\boldsymbol{\theta}}, D)$ | $R_{\mathrm{adv}}(f_{\boldsymbol{\theta}}, D^+_{f_{\boldsymbol{\theta}}})$ | $R_{\mathrm{tf}}(f_{\boldsymbol{\theta}}, D^{\mathrm{cnr}}_{f_{\boldsymbol{\theta}}})$ |
| 1 | 8.15 | 47.96 | 23.27 | 89.38 | 58.05 |
| 5 | 13.53 | 56.46 | 25.07 | 81.94 | 69.92 |
| 10 | 20.75 | 55.10 | 28.09 | 71.14 | 67.14 |
| 20 | 24.95 | 52.75 | 31.58 | 63.53 | 63.95 |
| 40 | 26.13 | 50.66 | 34.86 | 59.89 | 62.88 |
| 60 | 26.51 | 49.56 | 36.99 | 57.93 | 63.15 |
| 80 | 25.91 | 47.09 | 38.70 | 57.73 | 59.85 |
| 100 | 25.98 | 46.16 | 39.51 | 57.05 | 58.48 |

Table 14: Experiments of ResNet-18 with different robustness (%) on CIFAR-10. Models were trained on Quality data by PGD-AT with different $\epsilon$ under the $\ell_\infty$ threat model.

| $\epsilon$ | Robustness | | Risk | | |
|---|---|---|---|---|---|
| | Clean | Two-faced | $R_{\mathrm{nat}}(f_{\boldsymbol{\theta}}, D)$ | $R_{\mathrm{adv}}(f_{\boldsymbol{\theta}}, D^+_{f_{\boldsymbol{\theta}}})$ | $R_{\mathrm{tf}}(f_{\boldsymbol{\theta}}, D^{\mathrm{cnr}}_{f_{\boldsymbol{\theta}}})$ |
| 8/255 | 28.85 | 55.31 | 43.49 | 48.94 | 95.66 |
| 12/255 | 30.53 | 51.25 | 48.38 | 40.85 | 98.24 |
| 14/255 | 30.10 | 46.68 | 52.95 | 36.02 | 97.82 |
| 16/255 | 29.20 | 42.85 | 56.93 | 32.20 | 98.41 |
| 18/255 | 27.59 | 38.32 | 61.52 | 28.30 | 98.53 |
| 20/255 | 26.42 | 35.26 | 64.68 | 25.19 | 99.32 |

# D  Experiments on Naturally Trained Models

## D.1  Omitted Tables

Table 15: Robustness (%) of naturally trained VGG-16 and WideResNet-28-10 with $\ell_\infty$ threat model on CIFAR-10.

| Dataset | Verification Examples | VGG-16 | | WideResNet-28-10 | |
|---|---|---|---|---|---|
| | | Poisoning | Quality | Poisoning | Quality |
| CIFAR-10 | Clean | 0.00 | 0.00 | 0.00 | 0.00 |
| | Hypocritical | **1.04** | 0.05 | 0.00 | 0.00 |
| | Two-faced | 0.00 | **0.08** | 0.00 | **0.01** |

Table 16: Robustness (%) of naturally trained models with $\ell_\infty$ threat model on different datasets.

| Dataset | Verification Examples | MLP | | ResNet-18 | |
|---|---|---|---|---|---|
| | | Poisoning | Quality | Poisoning | Quality |
| CIFAR-10 | Clean | 0.51 | 0.79 | 0.00 | 0.00 |
| | Hypocritical | 29.67 | 32.81 | 0.00 | 0.00 |
| | Two-faced | **33.49** | **36.53** | 0.00 | **0.02** |
| SVHN | Clean | 0.00 | 2.96 | 0.00 | 0.60 |
| | Hypocritical | 0.08 | **41.59** | 0.00 | 1.14 |
| | Two-faced | **6.20** | 36.52 | 0.00 | **11.66** |
| CIFAR-100 | Clean | 0.07 | 1.32 | 0.00 | 0.01 |
| | Hypocritical | 3.67 | 16.85 | **0.07** | **0.17** |
| | Two-faced | **8.27** | **20.68** | 0.00 | **0.17** |
| Tiny-ImageNet | Clean | 0.00 | 0.07 | 0.00 | 0.00 |
| | Hypocritical | 0.71 | 4.91 | **0.16** | 0.67 |
| | Two-faced | **1.43** | **7.31** | 0.08 | **0.93** |

