# OpenReview forum: "On the Vulnerability of Adversarially Trained Models Against Two-faced Attacks"
_ICLR.cc/2024/Conference — ICLR 2024 poster_

### Official Review · Reviewer_ox78 · 2023-10-30

**Soundness:** 4 excellent
**Presentation:** 3 good
**Contribution:** 3 good
**Rating:** 6
**Confidence:** 5

**Summary:**

This work described the problem of hypocritical examples which added tiny perturbations to clean test inputs to increase the targeted model's performance. These samples can mislead machine learning practitioners into assuming that their machine learning models are good enough to be applied in the real world. The authors also mentioned that these samples can be crafted from adversarial examples to provide a false sense of adversarial robustness. They called this two-faced attack and provided its problem formulation and algorithm to create such an attack. They proposed a countermeasure by increasing the perturbation bound of adversarial training to get the best tradeoff between adversarial risk and two-faced risk. After that, they showed a bunch of experiment with several datasets.

**Strengths:**

- The paper has a strong formulation and background of hypocritical examples on adversarial examples.
- The experiment covers several kinds of datasets.
- The literature review is good and updated.
- The paper is well-organized and easy to follow.

**Weaknesses:**

- The background and motivation are not convincing to me. The authors may make it more motivating, and probably, look at the previous work (Tao, 2022b) since it is very convincing.
- The experiments for the countermeasure (enlarging the budget) are limited. The authors may need to add more experiments for it.
- The contribution of this paper is limited because the content is very similar to the previous work (Tao, 2022b), but changes from clean test samples to adversarial test samples. The authors may focus more on the countermeasure.

**Questions:**

- In Appendix A, I do know how you derive from E[1(f(x'') = y) * 1(f(x_{adv} \neq y)] to E[1(f_{adv} \neq y)] * E[1(f(x'') = y)]. It is in the lines 7 and 8 if you count the line R_{rhpy}(f,D) as line 1.

---

> ### Author Response · Authors · 2023-11-15
> **Reply to Reviewer ox78 (Part 1/2)**
>
> We thank the reviewer for the valuable and constructive comments, which provide considerably helpful guidance to improve the quality of our paper. Our point-by-point responses to the reviewer's mentioned questions (Q) and weaknesses (W) are provided as follows.
>
> **Q1: In Appendix A, I do not know how you derive from E[1(f(x'') = y) * 1(f(x_{adv} \neq y)] to E[1(f_{adv} \neq y)] * E[1(f(x'') = y)]. It is in the lines 7 and 8 if you count the line R_{rhpy}(f,D) as line 1.**
>
> Thanks so much for raising this point, which helped us identify an oversight in our derivation process and avoid the omission of a component in the final result. Fortunately, this has no impact on our conclusions. We have made corrections and uploaded the revised version. Thanks again for the reviewer's carefulness!
>
> **W1: The background and motivation are not convincing to me.**
>
> We have further enriched our background and motivation, and highlighted these changes in the introduction section of the paper. These can be summarized as follows.
>
> Due to the high probability of failures of model deployment in practical applications, it is necessary for practitioners to conduct pre-deployment verification and post-deployment long-term verification to enhance the success rate of deployment [1], especially in safety-critical tasks. Therefore, this work explores the existence of verification-time threats that hypocritically fool the robustly trained models by slightly perturbing the verification data.
>
> [1] Andrei Paleyes, Raoul-Gabriel Urma, and Neil D. Lawrence. Challenges in deploying machine learning: A survey of case studies. ACM Computer Survey, 55(6), 2022.
>
> **W2: The experiments for the countermeasure (enlarging the budget) are limited. The authors may need to add more experiments for it.**
>
> Thanks for the suggestion. We have conducted additional experiments to supplement the experimental results.
>
> 1) We trained ResNet-18 on CIFAR-100, using PGD-AT with different budgets under $\ell_{\infty}$ norm:
>
> | $\epsilon$ | Clean |   $\ $ TF    | $R_{\text{nat}}(f_{\theta}, D)$ | $R_{\text{adv}}(f_{\theta}, D^{+})$ | $R_{\text{tf}}(f_{\theta}, D^{\text{cnr}})$ |
> | ------- | :-----: | :-----: | :-----: | :------: | :--------: |
> | 8/255  | 28.85 | 55.31 | 43.49 | 48.94 | 95.66 |
> | 12/255 | 30.53 | 51.25 | 48.38 | 40.85 | 98.24 |
> | 14/255 | 30.10 | 46.68 | 52.95 | 36.02 | 97.82 |
> | 16/255 | 29.20 | 42.85 | 56.93 | 32.20 | 98.41 |
> | 18/255 | 27.59 | 38.32 | 61.52 | 28.30 | 98.53 |
> | 20/255 | 26.42 | 35.26 | 64.68 | 25.19 | 99.32 |
>
> 2) We trained ResNet-18 on CIFAR-10, using THRM [1] with different $\lambda$ under $\ell_{\infty}$ norm:
>
> | $\lambda$ | Clean |   $\ $ TF    | $R_{\text{nat}}(f_{\theta}, D)$ | $R_{\text{adv}}(f_{\theta}, D^{+})$ | $R_{\text{tf}}(f_{\theta}, D^{\text{cnr}})$ |
> | ------- | :-----: | :-----: | :-----: | :------: | :--------: |
> | 1    | 4.53  | 39.55 | 5.56  | 95.20 | 38.95 |
> | 5    | 23.87 | 75.54 | 8.75  | 73.84 | 76.68 |
> | 10   | 32.75 | 79.5  | 11.83 | 62.85 | 84.35 |
> | 20   | 35.85 | 73.14 | 9.79  | 60.25 | 68.59 |
> | 40   | 43.22 | 76.03 | 20.59 | 45.57 | 90.66 |
> | 60   | 43.66 | 73.02 | 24.56 | 42.12 | 92.38 |
> | 80   | 41.92 | 71.98 | 25.22 | 43.94 | 91.47 |
> | 100  | 43.9  | 69.51 | 29.08 | 38.09 | 94.78 |
>
> 3) We trained ResNet-18 on CIFAR-100, using THRM with different $\lambda$ under $\ell_{\infty}$ norm:
>
> | $\lambda$ | Clean |   $\ $ TF    | $R_{\text{nat}}(f_{\theta}, D)$ | $R_{\text{adv}}(f_{\theta}, D^{+})$ | $R_{\text{tf}}(f_{\theta}, D^{\text{cnr}})$ |
> | ------- | :-----: | :-----: | :-----: | :------: | :--------: |
> | 1    | 8.15 | 47.96 | 23.27 | 89.38 | 58.05 |
> | 5    | 13.53 | 56.46 | 25.07  | 81.94 | 69.92 |
> | 10   | 20.75 | 55.10 | 28.09 | 71.14 | 67.14 |
> | 20   | 24.95 | 52.75 | 31.58 | 63.53 | 63.95 |
> | 40   | 26.13 | 50.66 | 34.86 | 59.89 | 62.88 |
> | 60   | 26.51 | 49.56 | 36.99 | 57.93 | 63.15 |
> | 80   | 25.91 | 47.09 | 38.70 | 57.73 | 59.85 |
> | 100  | 25.98 | 46.16 | 39.51 | 57.05 | 58.48 |
>
> [1] Lue Tao, Lei Feng, Jinfeng Yi, and Songcan Chen. With false friends like these, who can notice mistakes? In AAAI, 2022b.
>
> Continue by the next post.

---

> ### Author Response · Authors · 2023-11-15
> **Reply to Reviewer ox78 (Part 2/2)**
>
> **W3: The contribution of this paper is limited because the content is very similar to the previous work (Tao, 2022b), but changes from clean test samples to adversarial test samples. The authors may focus more on the countermeasure.**
>
> As you mentioned, the previous work (Tao, 2022b) focused on **accuracy**, while our work focuses on **robustness** during the verification phase. Hence there is a clear difference, which makes them two entirely different tasks. Our contributions to the task of two-faced attacks are completely new. In addition, we have made some new findings regarding two-faced attacks:
>
> 1) Models with different architectures and adversarial training methods are susceptible to two-faced attacks, leading to falsely elevated verification robustness;
> 2) Models with higher adversarial robustness tend to be more vulnerable to two-faced attacks.
>
> Therefore, two-faced attacks are pervasive and tricky, and cannot be simply ignored.
>
> In terms of countermeasures, we have included more experimental results:
> 1) Experimental results of PGD-AT with different budgets under $\ell_{\infty}$ norm on CIFAR-100;
> 2) Experimental results of THRM with different $\lambda$ under $\ell_{\infty}$ norm on CIFAR-10;
> 3) Experimental results of THRM with different $\lambda$ under $\ell_{\infty}$ norm on CIFAR-100.
>
> We presented the relevant results in W2 and have added them to the appendix of the paper.
>
> We hope that our responses and the updated experiments have addressed your concerns. We are always willing to discuss and address any further questions.

---

> ### Author Response · Authors · 2023-11-20
> **Kind reminder to Reviewer ox78**
>
> Dear Reviewer ox78,
>
> Sorry to disturb you. We sincerely appreciate your valuable comments. We understand that you may be too busy to check our rebuttal. We would like to further provide brief answers here to the issues that may be your primary concerns.
>
> We appreciate your inquiries regarding the formula derivations, which helped us to identify an oversight in our derivation process. Fortunately, this has no impact on our conclusions.
>
> In addition, for the countermeasures, we have supplemented more experiments, including experiments on different datasets and experiments on the THRM defense strategy. We also discussed the possibility of incorporating two-faced samples into adversarial training methods as countermeasures.
>
> Finally, our contributions significantly differ from the previous work (Tao, 2022b) that focused on the **accuracy**. We concentrate on the **robustness** in the verification phase. Our contributions to the task of two-faced attacks are completely new. In addition, we have made some new findings regarding two-faced attacks:
>
> 1) Models with different architectures and adversarial training methods are susceptible to two-faced attacks, leading to falsely elevated verification robustness;
> 2) Models with higher adversarial robustness tend to be more vulnerable to two-faced attacks.
>
> We sincerely hope that the above answers can address your concerns. We look forward to your response and are willing to answer any questions.

---

> > ### Comment · Reviewer_ox78 · 2023-11-21
> >
> > I see that you revised the paper according to my concern and clearly explained your contribution. Also, you explained the derivation in the appendix. Therefore, I raised the score for you. Thanks for taking my comments into your consideration. Please continue producing good works!

---

> > > ### Author Response · Authors · 2023-11-21
> > >
> > > Thank you so much for acknowledging our contributions! We really appreciate your valuable comments and constructive suggestions, which definitely have helped us to improve the quality of our paper!

---

### Official Review · Reviewer_J5b2 · 2023-10-30

**Soundness:** 4 excellent
**Presentation:** 3 good
**Contribution:** 4 excellent
**Rating:** 8
**Confidence:** 4

**Summary:**

This paper shows that adversarially trained models are vulnerable to a new threat called two-faced
attacks, where slight perturbations in input features are crafted to make the model exhibit a false sense
of robustness in the verification phase. This paper also shows that this threat is pervasive and tricky,
because many types of models suffer from this threat, and models with higher adversarial robustness
tend to be more vulnerable. Besides, this paper gives a formal formulation for this threat and discloses
its relationship with adversarial risk. This paper also proposes a simple countermeasure to circumvent
the threat. Empirical results have validated the arguments presented in this paper.

**Strengths:**

1. It is important to disclose the existence of such two-faced attacks in the model verification phase, as
deploying substandard models (with low adversarial robustness) in reality could cause serious security
issues.

2. There are some interesting findings that can demonstrate the practical importance of two-faced
attacks. For example, many types of models suffer from this threat, and models with higher
adversarial robustness tend to be more vulnerable.

3. This paper mathematically formulates the two-faced attacks and the two-faced risk, and provide a
theoretical analysis on its relationship with adversarial risk, and provide a discussion on possible
countermeasures to circumvent two-faced attacks.

4. Experimental results are supportive.

**Weaknesses:**

The two-faced attacks are the key of this paper, but the realistic application domains are still unclear.

From Figure 1(a), this paper gives a machine learning workflow that shows the adversarial robustness
can be affected by the two-faced attacks in the model verification. However, this paper did not
provide a real-world example that the two-faced attacks can be applied into and can cause serious
security issues. This point can further strengthen the significance of the two-faced attacks.

**Questions:**

Can the authors provide a real-world example that the two-faced attacks can be applied into?

Can the authors provide more potential countermeasures against two-faced attacks, apart from the
ones mentioned in Section 3.3?

Other problems please see **Weaknesses.**

Overall, I like the idea and analysis in this paper.  I expect the problems could be clarified and addressed in the rebuttal.

---

> ### Author Response · Authors · 2023-11-15
> **Reply to Reviewer J5b2 (Part 1/2)**
>
> Thank you for your precious time in reviewing our work. Our point-by-point responses to the reviewer's mentioned questions (Q) and weaknesses (W) are provided as follows.
>
> **Q1: Can the authors provide a real-world example that the two-faced attacks can be applied into?**
>
> According to the statistics in practical deployment [1], 85\% of attempted deployments ultimately fail to achieve the intended results in the production environment. Validation is one of the keys to a successful deployment [2]. A practical real-world example is computer virus detection. There is a possibility that the collected verification data used to validate the virus detection software has been already influenced by attackers using two-faced attacks, which would lead to falsely elevated verification results. In this case, the deployment of such a virus detection software may result in serious consequences.
>
> [1] https://web.archive.org/web/20220707220351/https://www.decisioniq.com/blog/ai-project-failure-rates-are-high
>
> [2] Andrei Paleyes, Raoul-Gabriel Urma, and Neil D. Lawrence. Challenges in deploying machine learning: A survey of case studies. ACM Computer Survey, 55(6), 2022.
>
> **Q2: Can the authors provide more potential countermeasures against two-faced attacks, apart from the ones mentioned in Section 3.3?**
>
> In the submission, the main focus is on revealing the threat of two-faced attacks for the first time, while a mitigation strategy by increasing the defense budget was presented.
> In addition, we experiment with THRM [1] as another mitigation strategy. Specifically, we train ResNet-18 with THRM using different $\lambda$ under the $\ell_{\infty}$ norm and $\epsilon=8/255$ on CIFAR-10 and CIFAR-100. The experimental results are as follows:
> 1) The experiment of THRM with different $\lambda$ under $\ell_{\infty}$ norm and $\epsilon=8/255$ against two-faced attacks on CIFAR-10:
>
> | $\lambda$ | Clean |   $\ $ TF    | $R_{\text{nat}}(f_{\theta}, D)$ | $R_{\text{adv}}(f_{\theta}, D^{+})$ | $R_{\text{tf}}(f_{\theta}, D^{\text{cnr}})$ |
> | ------- | :-----: | :-----: | :-----: | :------: | :--------: |
> | 1    | 4.53  | 39.55 | 5.56  | 95.20 | 38.95 |
> | 5    | 23.87 | 75.54 | 8.75  | 73.84 | 76.68 |
> | 10   | 32.75 | 79.5  | 11.83 | 62.85 | 84.35 |
> | 20   | 35.85 | 73.14 | 9.79  | 60.25 | 68.59 |
> | 40   | 43.22 | 76.03 | 20.59 | 45.57 | 90.66 |
> | 60   | 43.66 | 73.02 | 24.56 | 42.12 | 92.38 |
> | 80   | 41.92 | 71.98 | 25.22 | 43.94 | 91.47 |
> | 100  | 43.9  | 69.51 | 29.08 | 38.09 | 94.78 |
>
> 2) The experiment of THRM with different $\lambda$ under $\ell_{\infty}$ norm and $\epsilon=8/255$ against two-faced attacks on CIFAR-100:
>
> | $\lambda$ | Clean |   $\ $ TF    | $R_{\text{nat}}(f_{\theta}, D)$ | $R_{\text{adv}}(f_{\theta}, D^{+})$ | $R_{\text{tf}}(f_{\theta}, D^{\text{cnr}})$ |
> | ------- | :-----: | :-----: | :-----: | :------: | :--------: |
> | 1    | 8.15 | 47.96 | 23.27 | 89.38 | 58.05 |
> | 5    | 13.53 | 56.46 | 25.07  | 81.94 | 69.92 |
> | 10   | 20.75 | 55.10 | 28.09 | 71.14 | 67.14 |
> | 20   | 24.95 | 52.75 | 31.58 | 63.53 | 63.95 |
> | 40   | 26.13 | 50.66 | 34.86 | 59.89 | 62.88 |
> | 60   | 26.51 | 49.56 | 36.99 | 57.93 | 63.15 |
> | 80   | 25.91 | 47.09 | 38.70 | 57.73 | 59.85 |
> | 100  | 25.98 | 46.16 | 39.51 | 57.05 | 58.48 |
>
> It can be seen from the above experimental results that the aforementioned strategies partially mitigate the risk of two-faced attacks, but such a risk remains significant and cannot be ignored. We have updated these experimental results in the appendix.
>
> Furthermore, we can incorporate our two-faced examples (derived from the two-faced risk) into existing adversarial training frameworks, such as TRADES and MART. In these frameworks, KL divergence is used to measure the difference in the output distributions between clean and adversarial examples. We can simply replace the adversarial examples in the KL loss term with our two-faced examples.
>
> However, because of the iterative generation of two-faced examples in a bi-level optimization formulation, the whole optimization process (i.e., alternately generating two-faced examples and conducting adversarial training) is highly time-consuming. Hence, we did not adopt this countermeasure. We will explore potential methods to improve the optimization efficiency for incorporating our generation of two-faced examples into existing adversarial training frameworks, in future work.
>
> [1] Lue Tao, Lei Feng, Jinfeng Yi, and Songcan Chen. With false friends like these, who can notice mistakes? In AAAI, 2022b.
>
> Continue by the next post.

---

> ### Author Response · Authors · 2023-11-15
> **Reply to Reviewer J5b2 (Part 2/2)**
>
> **W1: The two-faced attacks are the key of this paper, but the realistic application domains are still unclear.**
>
> Two-faced attacks focus on the robustness during the model verification phase. The verification phase is significant, especially in security-critical applications such as autonomous driving or computer virus detection. Additionally, considering the need for long-term monitoring and updates during the extended deployment of models, two-faced attacks could be applicable in this phase as well. Due to the high cost of detection and update, researchers may be deceived not to update the problematic model in the face of long-term false high validation results, which may eventually lead to a series of security problems.
>
> We hope that our responses can satisfactorily address your concerns. Thank you again for your time to provide us with valuable feedback.

---

> > ### Comment · Reviewer_J5b2 · 2023-11-21
> > **Thanks for the reply.**
> >
> > Thanks for your reply on my questions.
> >
> > This paper studies two-faced attack when adversarially training models. In the rebuttal, I'm happy that the authors provide a real scenario about the application of it. Meanwhile, the new experiments show the effect of the mitigation strategy, which is helpful to understand the two-faced attacks.
> >
> > Therefore, I think this paper did a good job in the field of adversarial robustness. I'd love to maintain my score 8.

---

> > > ### Author Response · Authors · 2023-11-21
> > >
> > > Thank you so much for your positive and insightful feedback! We sincerely appreciate your valuable time on our paper!

---

### Official Review · Reviewer_rs75 · 2023-11-03

**Soundness:** 3 good
**Presentation:** 3 good
**Contribution:** 3 good
**Rating:** 8
**Confidence:** 5

**Summary:**

The vulnerability in machine learning models that this paper reveals is called "two-faced attacks," wherein small input perturbations trick the model during verification and give the impression that it has high adversarial robustness. This observation raises concerns since it may result in security vulnerabilities if models that are not reliable are used based on false robustness evaluations. The phenomena is common and, surprisingly, more prevalent in models that are thought to be quite robust. The authors provide a framework for comprehending this problem and make some initial recommendations for solutions. Rather than accepting adversarial robustness at face value, they advise individuals to evaluate it rigorously.

**Strengths:**

This study takes a leading role in exposing the vulnerability of adversarially trained models to 'two-faced attacks', which fraudulently overestimate the robustness of the model during the verification stage.
Notable features include its comprehensive evaluation across a variety of model architectures, datasets, and training techniques, as well as its introduction of the 'two-faced risk' notion to reconcile theory and empirical results.

Also, the work represents a challenging problem in adversarial defense, as it reveals a counterintuitive trade-off: models that are more resilient to adversarial examples are also more vulnerable to two-faced attacks.

However, the paper's implications for practice are highlighted by a strong need for rigorous validation processes prior to deployment, particularly in safety-critical domains. This call urges a reevaluation of how adversarial robustness is measured and perceived in the field.

**Weaknesses:**

Although the study identified two-faced attacks against adversarially trained models in a novel way, it might not provide practitioners with sufficient defenses to address these weaknesses. Though interesting, its theoretical investigation of two-faced risk may prove difficult to implement in real-world scenarios and may not provide enough information about mitigation strategies.

Additionally, there may be an overemphasis on two-faced attacks, which might mask other important security issues that need to be taken seriously. Furthermore, a lack of clarity in the procedures for experiments may make them difficult to repeat and inhibit future studies.

Finally, there can be a gap in comprehensive risk management techniques if the paper does not include a comprehensive explanation of how to balance adversarial risks, such as two-faced risk, against other risks.

**Questions:**

Just two questions:

Is it possible to include the idea of two-faced risk into current adversarial training frameworks without making major changes?

How can long-term model validation procedures be affected by the future evolution of two-faced attacks?


Finally, two suggestions:

Future work should focus on developing more comprehensive defense strategies against two-faced attacks that can be easily implemented in real-world systems.

Perform research on how models might eventually be exposed to fraudulent attacks, especially if adversaries and attack techniques advance in sophistication.

---

> ### Author Response · Authors · 2023-11-15
> **Reply to Reviewer rs75 (Part 1/3)**
>
> Thanks for the valuable and encouraging comments! We are so grateful to see that our carefully prepared work can be recognized by the reviewer. Our point-by-point responses to the reviewer's mentioned questions (Q) and weaknesses (W) are provided as follows.
>
> **Q1: Is it possible to include the idea of two-faced risk into current adversarial training frameworks without making major changes?**
>
> Yes, we can incorporate our two-faced examples (derived from the two-faced risk) into existing adversarial training frameworks, such as TRADES and MART. In these algorithms, KL divergence is used to measure the difference in output distributions between clean and adversarial examples. We can simply replace the adversarial examples in the KL divergence term with our two-faced examples.
>
> However, because of the iterative generation of two-faced examples in a bi-level optimization formulation, the whole optimization process (i.e., alternately generating two-faced examples and conducting adversarial training) is highly time-consuming. Hence, we did not adopt this countermeasure. We will explore potential methods to improve the optimization efficiency for incorporating our generation of two-faced examples into existing adversarial training frameworks, in future work.
>
> **Q2: How can long-term model validation procedures be affected by the future evolution of two-faced attacks?**
>
> If the attacker persists in conducting two-faced attacks, the long-term model verification procedures may also exhibit falsely elevated performance. Since long-term validation and model updates are challenging and time-consuming, researchers may not update their models if they observe falsely elevated validation metrics for a long time, opting instead to continue deploying a potentially problematic model. This could ultimately lead to a cascade of security issues. This is also why we emphasize in the paper not to overly trust these validation results.
>
> **W1: Though interesting, its theoretical investigation of two-faced risk may prove difficult to implement in real-world scenarios.**
>
> We would like to clarify that, for applications in the real digital world, our two-faced attacks are easily applicable. As long as researchers collect verification data from the online world, there is a risk of being vulnerable to two-faced attacks. For applications in the physical world, e.g., autonomous driving, two-faced attacks are feasible by combining them with techniques such as adversarial patches. Overall, two-faced attacks are a real threat to AI if adversaries are aware of where the validation data is collected.
>
> Continue by the next post.

---

> ### Author Response · Authors · 2023-11-15
> **Reply to Reviewer rs75 (Part 2/3)**
>
> **W2: May not provide enough information about mitigation strategies.**
>
> In this paper, our primary focus is for the first time to reveal the threat of two-faced attacks. Regarding mitigation strategies, in addition to increasing the budget as mentioned in the paper, we also explored the mitigation capability of THRM [1] with different $\lambda$ under $\ell_{\infty}$ norm and $\epsilon=8/255$ against two-faced attacks on CIFAR-10:
> | $\lambda$ | Clean |   $\ $ TF    | $R_{\text{nat}}(f_{\theta}, D)$ | $R_{\text{adv}}(f_{\theta}, D^{+})$ | $R_{\text{tf}}(f_{\theta}, D^{\text{cnr}})$ |
> | ------- | :-----: | :-----: | :-----: | :------: | :--------: |
> | 1    | 4.53  | 39.55 | 5.56  | 95.20 | 38.95 |
> | 5    | 23.87 | 75.54 | 8.75  | 73.84 | 76.68 |
> | 10   | 32.75 | 79.5  | 11.83 | 62.85 | 84.35 |
> | 20   | 35.85 | 73.14 | 9.79  | 60.25 | 68.59 |
> | 40   | 43.22 | 76.03 | 20.59 | 45.57 | 90.66 |
> | 60   | 43.66 | 73.02 | 24.56 | 42.12 | 92.38 |
> | 80   | 41.92 | 71.98 | 25.22 | 43.94 | 91.47 |
> | 100  | 43.9  | 69.51 | 29.08 | 38.09 | 94.78 |
>
> The experiment of THRM with different $\lambda$ under $\ell_{\infty}$ norm and $\epsilon=8/255$ against two-faced attacks on CIFAR-100:
> | $\lambda$ | Clean |   $\ $ TF    | $R_{\text{nat}}(f_{\theta}, D)$ | $R_{\text{adv}}(f_{\theta}, D^{+})$ | $R_{\text{tf}}(f_{\theta}, D^{\text{cnr}})$ |
> | ------- | :-----: | :-----: | :-----: | :------: | :--------: |
> | 1    | 8.15 | 47.96 | 23.27 | 89.38 | 58.05 |
> | 5    | 13.53 | 56.46 | 25.07  | 81.94 | 69.92 |
> | 10   | 20.75 | 55.10 | 28.09 | 71.14 | 67.14 |
> | 20   | 24.95 | 52.75 | 31.58 | 63.53 | 63.95 |
> | 40   | 26.13 | 50.66 | 34.86 | 59.89 | 62.88 |
> | 60   | 26.51 | 49.56 | 36.99 | 57.93 | 63.15 |
> | 80   | 25.91 | 47.09 | 38.70 | 57.73 | 59.85 |
> | 100  | 25.98 | 46.16 | 39.51 | 57.05 | 58.48 |
>
> For the experiment of increasing the budget for PGD-AT, we further extended it to the CIFAR-100 dataset:
> | $\epsilon$ | Clean |   $\ $ TF    | $R_{\text{nat}}(f_{\theta}, D)$ | $R_{\text{adv}}(f_{\theta}, D^{+})$ | $R_{\text{tf}}(f_{\theta}, D^{\text{cnr}})$ |
> | ------- | :-----: | :-----: | :-----: | :------: | :--------: |
> | 8/255  | 28.85 | 55.31 | 43.49 | 48.94 | 95.66 |
> | 12/255 | 30.53 | 51.25 | 48.38 | 40.85 | 98.24 |
> | 14/255 | 30.10 | 46.68 | 52.95 | 36.02 | 97.82 |
> | 16/255 | 29.20 | 42.85 | 56.93 | 32.20 | 98.41 |
> | 18/255 | 27.59 | 38.32 | 61.52 | 28.30 | 98.53 |
> | 20/255 | 26.42 | 35.26 | 64.68 | 25.19 | 99.32 |
>
> From the above experimental results, it can be observed that the aforementioned strategies partially mitigate the impact of two-faced attacks, but such a risk remains significant and cannot be ignored. We have updated these experimental results in the appendix.
>
> In addition, for other potential mitigation strategies, as mentioned in Q1, we can integrate $x_{\text{tf}}$ into certain existing adversarial training frameworks.
>
> [1] Lue Tao, Lei Feng, Jinfeng Yi, and Songcan Chen. With false friends like these, who can notice mistakes? In AAAI, 2022b.
>
> Continue by the next post.

---

> ### Author Response · Authors · 2023-11-15
> **Reply to Reviewer rs75 (Part 3/3)**
>
> **W3: There may be an overemphasis on two-faced attacks, which might mask other important security issues that need to be taken seriously.**
>
> We agree with the reviewer's perspective. The primary research objective of this paper is to uncover the threats posed by two-faced attacks, but it is undeniable that other significant security issues should not be overlooked. Balancing the risks associated with two-faced attacks and other security concerns is indeed a topic worthy of in-depth exploration. For instance, integrating the two-faced risk into existing adversarial training frameworks merits thorough investigation. However, as discussed in the response to Q1, the resulting optimization process (i.e., alternately generating two-faced examples and conducting adversarial training) is highly time-consuming, especially given vast training data. Therefore, there is a need for efficiency optimization to address this issue.
>
> **W4: A lack of clarity in the procedures for experiments may make them difficult to repeat and inhibit future studies.**
>
> Thanks for raising this concern. We have provided more information in the experimental procedure section in Appendix B and have uploaded the revised version.
>
> **W5: There can be a gap in comprehensive risk management techniques if the paper does not include a comprehensive explanation of how to balance adversarial risks, such as two-faced risk, against other risks.**
>
> We agree with the reviewer's feedback. The issue of balancing these risks is indeed worth investigating. While our primary focus in this paper is to uncover the threats posed by two-faced attacks, we presented a preliminary exploration of the trade-off between adversarial risk and two-faced risk in Figure 3 of the submission. As mentioned in the response to Q1, the resulting optimization process is computationally inefficient. Hence, we plan to delve into these challenges in future work, and we anticipate that the reviewer understands the complexities involved.
>
> We hope that our responses can satisfactorily address your concerns. Thank you again for your time to provide us with valuable feedback.

---

### Public Comment · ~Boris_Zhu1 · 2023-11-13
**Some questions**

Dear authors,

This paper is interesting, but I have some questions.

It may be not true that your equation (4) is a challenging bi-level optimization. I think for any adversarial attacks $\delta(x)$, $-\delta$ are always satisfied the solution of two-face attacks, i.e., your equation (4). What is the difference between anti-adversarial attacks and two-face attacks? Although alternative optimization may results in different $\delta$ compared with adversarial attacks.

Not only on the robust dataset, even on non-robust dataset ($D_{f_{\theta}}^{cnr}$), the proposed two-faced risk is alway $1$, as one can always find a $\delta=-t$ to erase the perturbation. I am very confusing on the meaning of TF risks.

Overall, using the same budget $\epsilon$ for both $\delta$ and $t$ in TF attacks is quite confusing, I hope the authors could make some clarifications.

---

> ### Author Response · Authors · 2023-11-15
> **Clarifications**
>
> Thank you for your interest in our work!
>
> We would like to clarify that $-\delta^{\star}$ may not necessarily be the optimal solution that satisfies the formulation. In Algorithm 1 of our paper, we show the optimization process for $x_{\text{tf}}$. In Line 3, when calculating the optimal value $t^{\star}$, due to the nonlinearity of neural networks, there may exist a solution that results in a smaller loss than $-\delta^{\star}$. The same rationale also holds for Line 4, when calculating the optimal value $\delta^{\star}$.
>
> Besides, the two-faced risk on the non-robust dataset ($D_{f_{\theta}}^{\text{cnr}}$) is not always equal to 1. When conducting adversarial attacks on  $x_{\text{tf}}$ generated from $D_{f_{\theta}}^{\text{cnr}}$, the attack algorithm may not produce a countervailing perturbation (i.e., make $x_{\text{tf}}$ recover the clean $\boldsymbol{x}$ after adding an adversarial perturbation).  This is because such a perturbation might not optimally lead to the misclassification. Due to the nonlinearity of neural networks, there may exist perturbations in other directions that lead to the misclassification of the perturbed $x_{\text{tf}}$. Therefore, the two-faced risk on $D_{f_{\theta}}^{\text{cnr}}$ is potentially less than $1$.
>
> We hope that our response can address your concerns. Thanks again for your attention to our work and for your comment.

---

### Meta-Review · Area_Chair_AJJ3 · 2023-12-12

**Metareview:**

This work explored "hypocritical examples," tiny perturbations added to test inputs that inflate targeted model performance, potentially misleading practitioners. These samples, crafted from adversarial examples, create a false sense of adversarial robustness. The authors propose a countermeasure by increasing the perturbation bound of adversarial training to balance adversarial and "two-faced" risks. Experiments across several datasets are presented.

Strengths: The paper is well-structured and easy to read. It provides a thorough literature review and mathematically defines two-faced attacks and their associated risk. The authors establish a theoretical connection between this risk and adversarial risk. The paper also explores potential countermeasures for mitigating two-faced attacks. The experimental findings corroborate the theoretical framework.

Weaknesses: The authors' response to Reviewer J5b2's question "Can the authors provide a real-world example that the two-faced attacks can be applied into?" is insufficient. The authors provide virus detection as an example, but do not provide any reference showing that neural nets are used in virus detection. In addition, the authors should provide more examples in the domain of computer vision, which is a field where neural networks are widely used and adversarial attacks are a known problem.

**Justification For Why Not Higher Score:**

The authors did not provide a sufficiently concrete example of a real-world application of two-faced attacks, which limits the impact of this paper.

**Justification For Why Not Lower Score:**

This paper introduces a new attack and a new dimension of robustness for neural models.

---

### Decision · Program_Chairs · 2024-01-16

Accept (poster)